# Structural Equation Modeling of Common Cognitive Abilities in Preschool-Aged Children Using WPPSI-IV and BRIEF-P

**DOI:** 10.3390/children9071089

**Published:** 2022-07-21

**Authors:** Sören Fiedler, Nina Krüger, Monika Daseking

**Affiliations:** 1Department of Educational Psychology, Helmut Schmidt University/University of the Federal Armed Forces, 22043 Hamburg, Germany; m.daseking@hsu-hh.de; 2Department Differential Psychology and Psychological Assessment, Institute of Psychology, University of Hamburg, 20146 Hamburg, Germany; nina.krueger@uni-hamburg.de

**Keywords:** intelligence, executive functioning, WPPSI-IV, BRIEF-P, children, preschoolers, confirmatory factor analyses, structural equation modeling, common method variance

## Abstract

Various studies have addressed the relationship between intelligence and executive functions (EF). There is widespread agreement that EF in preschool children is a unitary construct in which the subordinate factors of Updating, Inhibition, and Shifting are still undifferentiated and correlate moderately with a general factor of intelligence (g). The aim of this study is to investigate the common structural relationship between these two constructs using confirmatory factor analysis. Furthermore, we intend to close the gap of more daily life-associated executive functions and replicate findings in preschool-aged children. Data from a sample of N = 124 average developed children without severe impairments (aged 4 years 0 months–6 years 11 months) were analyzed using the data pool of the standardization and validation studies on the German Wechsler Preschool and Primary Scale of Intelligence—Fourth Edition. Additionally, Executive functions were assessed using a standardized parent-completed questionnaire (BRIEF-P) on their children’s everyday behavior. A second-order factor solution revealed that a model with a loading of the common factor of general intelligence (*g*-factor) onto the EF factor fits the data best. To specify possible method effects due to different sources of measurements, a latent method factor was generated. The results indicate a heterogeneous method effect and a decreasing factor loading from g on to EF while controlling for the method factor.

## 1. Introduction

Executive functions (EF) are a multidimensional construct. As such, a large number of models exist that attempt to represent the basic processes of EF from various perspectives [1]. Generally, EF have been demonstrated in studies to be higher, self-regulatory, cognitive processes that control thinking and action, and are associated with goal-oriented behavior [2]. Three factors can usually be identified as core competencies of EF: (I) Cognitive flexibility (“shifting”), which regulates changes in attention, tasks, and strategy; (II) Inhibitory control (“inhibition”) to suppress premature, dominant, and/or automated responses; and (III) The power of working memory (“updating”) to keep information in memory and update it while it is being processed [3,4,5]. Studies in children have shown that these three factors can already be determined as distinct dimensions, though there is some overlap [6]. However, the ability to distinguish between these three areas in childhood depends on the research method and the age of the sample used [7].

Studies using factor analysis within EF have not yet clarified whether it is a unitary or multidimensional construct [8,9]. Researchers have also identified further components of EF, described as “emotions”, “beliefs”, or “desires”, such as the experience of reward and punishment, regulation of one’s own social behavior, and decision-making involving emotional and personal interpretation, which are considered “hot components” [10]. This work focuses on the above-mentioned “cold” core components of EF, where the corresponding cognitive processes generally do not involve much emotional arousal and are relatively “mechanistic” or “logical” in nature, focusing on cognitive flexibility, inhibitory control, and working memory [11].

Baddeley [12] underlines the central role that EF play in working memory [13]. He considers working memory functions to be regulated by a “central executive” that controls visual and phonological memory as well as episodic memory [14]. This model also describes various attention regulation processes, including inhibitory control and cognitive flexibility (switching/shifting). Taking the structure of working memory as an integral part of EF implies other differentiable processes from each, such as the control of attention to specific memories, the preservation of working memory contents, and the updating and manipulating of contents (updating) [3,15]. 

The idea of EF as a “central organizer” is also supported by neuroanatomical studies, where adults that have sustained damage to the prefrontal cortex area show deficits in areas identified with EF, such as everyday action planning and regulation. Surprisingly, this was not associated with general deficits in intelligence [16]. Imaging studies also show that the three EF mentioned above are associated with the activation of the prefrontal cortex, as are the fluid intelligence components [17]. 

From a developmental perspective, the maturation processes of these areas are associated with an increase in synaptic density around the age of seven, which contributes to a rapidly increasing development and differentiation of both intelligence and EF [18]. Cattell [19] pointed out that the two components of intelligence g proposed by him (fluid intelligence = g_f_ and crystalline intelligence = g_c_) are not yet clearly distinguishable in early childhood, but g_f_ is considered innate and g_c_ acquired and g emerges from investment success of g_f_ into g_c_. It is assumed that differentiation of intelligence into its fluid and crystalline parts occurs earlier than differentiation within EF [9]. Furthermore, EF and intelligence share a common neurobiological basis. Both are associated with neural activity in a frontoparietal network but also have their own specific neural correlates [20]. Overall, the connection between EF and intelligence is undoubted, and not only with regard to the biological maturation processes. EF generally increase from preschool age through school age and into early adulthood [21,22]. Furthermore, it can be assumed that the three core EF develop in different periods in the transition between childhood and adolescence [8,21]. Inhibition develops rapidly from preschool age onwards and changes to a smaller extent in later childhood, whereas the other two core EF develop more gradually [8]. The factorial structure of EF differs, as has already been indicated, depending on the different age groups and measurements used [9]. For example, the BRIEF-P, which is also used in this study, leads to different factor results within one measurement method [23,24,25,26]. 

There are various close relations between EF and intelligence, and the definitions of the two constructs are often described as overlapping or even coinciding [9,18,27,28,29]. EF tasks also involve other cognitive abilities, including other executive and nonexecutive processes, such as intelligence [8,28,30]. Cognitive performance measures, such as the WPPSI-IV, also implicitly include aspects of executive functions within the cognitive function domains that are measured [31]. Since executive functions are well represented in the Cattell–Horn–Carroll (CHC) model, a recent theoretical framework for intelligence research, it is not surprising that complex tasks show references to both EF and intelligence [32]. This underlines once more a clear association between executive functions and intellectual abilities.

It is still unclear whether the connection between intelligence and EF is the same in children, as many studies that aimed at correlating the two have shown contradictory results [33]. The findings on the three EF suggest a differential relation between EF and intelligence, in particular with the intelligence factors g_f_ and g_c_ [28,29]. Particular importance is attached to the function of working memory [9,28,34]. Findings by Friedman and colleagues [1,28] suggest that both intelligence factors are highly associated with working memory, but this does not apply to inhibitory control and cognitive flexibility. 

On the construct level, studies have demonstrated a differential relation between EF and fluid intelligence (g_f_) in adults, with working memory being a profound indicator of intelligence performance [35]. Further results show high correlations between cognitive flexibility (shifting) and fluid intelligence (g_f_) [36]. From a developmental psychology perspective, these results can hardly be generalized to childhood and are not undisputed. On the other hand, some argue that a latent EF variable could be responsible for the commonality of g_f_ and g_c_ [9,28]. In their longitudinal study, Rahbari and Vaillancourt [29] examined the relationship between executive functions and fluid and crystalline components of intelligence in 2–5-year-old children. Even at this young age, working memory was shown to be the central EF, which is strongly related to both verbal and non-verbal areas of intelligence assessed within the WPPSI-III [29].

The findings of Brydges and colleagues [9], Daseking and colleagues [37], and Rahbari and Vaillancourt [29], which focus on the relationship between EF and fluid intelligence, indicate moderate relationships. Based on previous findings, it can therefore be assumed that there are already connections between the individual components of executive functions and cognitive performance in the sense of intelligence in preschool-aged children, at least in the low to moderate range [37,38]. Nevertheless, no published study to date has used confirmatory factor analysis (CFA) to assess the common structure of EF and intelligence in preschool-aged children in a cross-informant approach. 

Based on the current literature, it can be concluded that EF are closely linked to intelligence. However, there is no clarity regarding a common factorial structure of EF and intelligence, even in preschoolers. Research suggests that EF could be another primary factor in a comprehensive intelligence model. Although there are factor-analytical studies on the constructs WPPSI-IV [39,40,41] and BRIEF-P [42,43], they do not take them into account in a common structural model to explore the connections more deeply. Accordingly, the current work presents a joint investigation of the factorial structure of intelligence and EF, which are recorded as cognitive abilities within the WPPSI-IV and as deficits in daily EF behavior within the BRIEF-P. 

## 2. Materials and Methods

### 2.1. Participants

Data were collected as part of the standardization and validation studies on the German Wechsler Preschool and Primary Scale of Intelligence—Fourth Edition [39,40]. For a random subsample of N = 124 children (49 girls, 75 boys) aged 4 years 0 months–6 years 11 months (M = 62.7 months; SD = 9.96; min.: 48; max.: 84), parents’ assessments of their children’s behavior in terms of executive functions (BRIEF-P) were also recorded. The data were collected from April 2016 to November 2017 at several locations in Germany. The assessment took place mainly in the rooms of cooperating test centers, kindergartens, daycare centers, and preschool classes of primary schools. In addition to a questionnaire on the sociodemographic characteristics of the family, parents also answered further questions about deficits in EF in everyday situations. The sample’s sociodemographic characteristics can be found in Table 1. 

This sample is not statistically different from the total sample used for standardization. To identify a behavioral abnormality with regard to the EF, a value of 1.5 SD above the average of the BRIEF-P (T-value ≥ 65) was used as the corresponding criterion. These values in the respective scales and indices of the BRIEF-P are considered abnormal behaviors [3]. Regarding cognitive performance, values above or below the average were only found at the level of the overall intelligence quotient. For this, 1.0 SD from the sample mean was chosen as the criterion (IQ ≥ 115; IQ ≤ 85). An IQ value of 115 is considered above average, values between 85 and 115 are considered average, and a value between 70 and 84 indicates learning disabilities according to ICD-10 [44] (for details see Appendix A).

### 2.2. Measurement

The WPPSI-IV [39] provides a measurement of intellectual abilities for children aged 2 years and 6 months to 7 years and 7 months. Since this study is based on a version for older children (age group 4 years 0 months–7 years 7 months), the scale structure of the version for younger children (2 years 6 months–3 years 11 months) is not shown here [39]. In contrast to the previous version, the WPPSI-III [45], which is based on three factors within the hierarchical framework, the factorial structure of the WPPSI-IV now comprises one second-order factor and five first-order factors. This structure is reflected by the fullscale IQ (FSIQ) and five primary indexes: verbal comprehension index (VCI), visual spatial index (VSI), fluid reasoning index (FRI), working memory index (WMI), and processing speed index (PSI). The WPPSI-IV version for older children includes fifteen subtests, ten of which are used to assess the primary indices: Information (IN), similarities (SIs), block design (BD), object assembly (OA), matrix reasoning (MR), picture concepts (PCs), picture memory (PM), zoo locations (ZLs), bug search (BS), and cancelation (CA). Five of these tests were optional and were only used if additional information was needed.

In a recent study, two of these optional tests were included (receptive vocabulary and picture naming). Regarding the reliability for older children, the subtests for VCI, VSI, FRI, and WMI indicated sufficient to excellent internal consistency coefficients with Cronbach’s alpha ranging from 0.81 to 0.93. The test–retest reliability coefficients for the PSI subtests, ranging from 0.72 to 0.74, turned out to be lower but still sufficiently good [1]. Based on factor analytical findings, the test publishers concluded that the second-order, five-factor structure was the most suitable model solution for this age group [1]. The test authors used two separate second-order models (5a, 5b) for the factorial structure of the WPPSI-IV. The first model (5a) was based on the ten primary subtests that are necessary to form the primary indexes of the WPPSI-IV and, on the other hand, to constitute the fullscale IQ (FSIQ). This model includes ten subtests, five primary factors, and an overall value of intelligence (*g*-factor) as a secondary factor. The other model (5b) is based on all fifteen available subtests. In addition, the VCI is divided into two sub-factors; one represents broader expressive language skills (Subtest PN), the other receptive language performance (Subtest RV) with little or no requirement for expressive language. Due to the unavailability of some optional subtests in the present study (VO, CO and AC), the second model is used in a modified form. 

The Behavior Rating Inventory of Executive Function—Preschool Version (BRIEF-P) [42,43] is a parent-, caregiver-, or teacher-report measure of a preschooler’s everyday EF behavior, and it is intended for a broad age range of 2 years through 6 years 11 months. The inventory represents a multicomponent view of EF in which specific EF domains are defined based on the theoretical framework, clinical practice, and extant research literature [43]. The BRIEF-P consists of 63 items that are rated on a three-point scale (“never”, “sometimes”, or “often”), indicating whether the described behaviors have been problematic in the last six months. The questionnaire comprises the five subscales F, shift, emotional control, working memory, and plan/organize and the Global Executive Composite (GEC) as an overall score. These scales form a three-factor model: (a) inhibitory self-control index (ISCI), (b) flexibility index (FI), and (c) emergent metacognition index (EMI) [3]. The reliability is available as internal consistency with a Cronbach’s α of 0.75–0.89 at the scale level and 0.86–0.91 at the index level. Regarding the factorial structure, empirical studies have yielded inconsistent results [23,24,25,26]. None of these sample-dependent results achieved general validity, although some of the models showed good fit. Due to the high intercorrelation of the primary indices, in which single items are assigned to several indices, and since the Global Executive Composite can be calculated directly from the five clinical scales, the primary indices were not included in this study.

### 2.3. Data Analyses

We conducted our data analyses in the following three steps. First, we transformed the data, calculated descriptive statistics and performed frequency analyses to provide an overview of the sample. Second, we examined the fit of the measurement models. 

Third, we investigated the structural relationship between intelligence and EF utilizing a structural equation modelling approach. 

In the present study, CFAs were conducted using AMOS 25 [46] and SPSS 25 for data preparation and further statistical analyses. The latent variable approach of second-order CFA was used, a specific form of structure equation modeling (SEM) for the investigation of the common structure in an integrated model. SEM facilitates the simultaneous quantitative estimation of the interdependencies of manifest (directly assessable) and latent (not directly assessable) variables and takes into account error variances [47]. The intelligence factor and EF are primary latent factors and are hypothesized to be essentially explained by a latent second-order factor resembling Spearman’s *g*-factor.

Technically, intelligence, as well as EF, have to be conceptualized as reflective measurement models following a factor analytic approach, thus proposing that the latent variables having a causal influence on their associated manifest indicators [47]. Therefore, previous results of the research on the structure of the WPPSI-IV and the BRIEF-P were included.

The psychometric property of the proposed intelligence model has already been evaluated within a standardization and validation study [39]. In addition, principal component analysis based on the data of the German standardization and validation studies confirm a three-factor model of the BRIEF-P, consistent with the English version [42,43,48]. For the reasons mentioned above, in this study, a modified factor model of the clinical scales is taken into account in order to avoid multiple inclusion of individual items on the three composite indices of the BRIEF-P (ISCI, FI, EMI) [42].

Post hoc and within the same sample, the factor reliability of the primary indices (WPPSI-IV) and the clinical scales (BRIEF-P), the average variance extracted (AVE), and the indicator reliability for all included subtests were estimated.

#### 2.3.1. Data Transformation

For all subtests of the WPPSI-IV, standardized values ($\bar{x}$ = 10; SD = 3) were used. The BRIEF-P Scale scores are available as T-standardized values ($\bar{x}$ = 50; SD = 10). Moreover, all scores were z-standardized to ensure better comparability of the estimated parameters. Due to the use of standardized values, the requirement of metric scalation of data for CFA [49] is assumed.

#### 2.3.2. Outliers, Normality, and Missing Values

Deviations from univariate normality were significant but small and according to the optical inspection, there were only small and tolerable deviations from normality. Following the recommendation of West [50], skewness measures were only evaluated as substantial deviations from coefficients of magnitude > 2. Multivariate outliers were examined and eliminated based on the Mahalanobis distance [47,51]. Although the exclusion of multivariate outliers reduced kurtosis, the test for skewness remained significant (Appendix A). A few multivariate outliers were left in the sample in order to avoid a critical drop in the sample, since it can be assumed that their achieved values correspond to their true values. Out of 134 individuals, an adjusted sample of 124 children remained after the elimination of the outliers. To check the multivariate normal distribution, Madia’s curvature coefficient was used [52]. The critical ratio (C.R.) should not be greater than 2.57 according to moderate-conservative testing [47] and with C.R. = 1.11 the value is within the acceptable range (Appendix A).

The inclusion criteria include the complete data set regarding all parameters used in the analyses, such as age, gender and education and migration background of parents, completely test data of all 12 subtest of the WPPSI-IV (see above) and the clinical scales of the BRIEF-P.

Generally, the exclusion criteria were a confirmed intelligence impairment (ICD-10; F7) as well as missing values in the examined variables, presence of severe neurological disorders, profound developmental disorder. Furthermore, for the statistical aspects, including the C.R. of Madia´s curvature coefficient > 2.5 (for a comprehensive description of the study participation conditions, please refer to the WPPSI-IV manual) [39].

#### 2.3.3. Model Fits and Comparisons

First, we carried out a common first-order CFA (CFA_0), including the measurement models of all primary intelligence factors as well as EF. Second, a second-order model additionally including a superordinate *g*-factor was fitted and analyzed. This model was tested against two nested factor models with fixed relations of the *g*-factor to the common EF factor. In the first nested model (CFA_1a), the loading of the *g*-factor on the EF factor was fixed to zero. In the second model (CFA_1b), the loading of the *g*-factor on the EF factor was fixed to one. The same procedure was applied in testing a model with two nested subfactors of the Verbal Comprehension Index (VCI). Again, the loading of the *g*-factor onto the EF factor was fixed to zero (CFA_2a) or one (CFA_2b). 

The model fit was evaluated based on the χ^2^ statistic [53], the root mean squared error of approximation (RMSEA) [54], the comparative fit index (CFI) [55], and standardized root mean square residual (SRMR) [56]. Following the suggestion of MacCallum and colleagues [57], we considered a RMSEA below 0.05 as an indicator of good fit and an RMSEA value of 0.10 as the cut-off for a poor-fitting model. While CFI values of 0.95 or higher correspond to a good fit [56,58], we considered a value above 0.90 to be an appropriate minimum for model acceptance is ≥0.90 [59]. Regarding the SRMR; a value less than 0.08 is generally considered a good fit [56], whereas values smaller than or equal to 0.10 may be interpreted as acceptable [60]. 

Model comparisons between different hierarchical models (zero, first, and second-order), were based on parsimony-adjusted measures (parsimony normed fit index; PNFI), [61], the Akaike information criterion (AIC) and the Bayesian information criterion (BIC) [46,62], as well as the consistent Akaike information criterion [63] (CAIC) expected cross-validation index [64] (ECVI). According to Williams and Holahan [65], differences of 0.06 to 0.09 between the models indicate substantial differences in the PNFI. 

#### 2.3.4. Method Factor

Since differences between both instruments are likely due to the different survey methods, we estimated and statistically controlled for method effects by implemented a primary latent method factor into the best fitting model [66,67]. Additionally, models with different equality restrictions for the cross-loadings on the method factor were specified: (1) model with indicator/trait factors only, (2) model with method factor only, (3) model with indicator/trait and method factor [67]. In the case of a method influence, we examined whether this affected the regression parameters in the structural equation model. For this purpose, the factor loadings of the method factor on the trait indicators were freely estimated (for specification, see Appendix A). Finally, a comparison between the estimated parameters of the model with and without the method factor are reported [67].

## 3. Results

Descriptive statistics of the manifest indicators for the sample are shown in Appendix A. The frequency analysis of the distribution of the recognizable values in the GEC (T-scores ≥ 65) showed percentages of 4–9% for the BRIEF-P scales (see corresponding column in Appendix A). Only eight children (6.5%) showed an abnormal value in the Global Executive Composite (GEC) of the BRIEF-P. 

### Structural Relation of EF and Intelligence

The modified second-order model was compared to two nested models, which both include the modifications mentioned above. Additionally, the *g*-factor loading onto the EF factor was fixed to zero in the first nested models (CFA_1a/CFA_2a) and fixed to one in the second models (CFA_1b/CFA_2b). An overview of the fit indices of compared nested models can be found in Table 2 A Likelihood-ratio test revealed a significantly better fit of CFA_1b compared to CFA_1a (Δχ²(1) = 16.91, *p* < 0.001). Additionally, the likelihood-ratio test revealed a significantly better fit of CFA_2b compared to CFA_2a (Δχ²(1) = 18.16, *p* < 0.001). The comparison between CFA_0 and CFA_1b as well as the comparison between CFA_1b and CFA_2b remained insignificant in a nested model approach. This is because dimensionality and higher order hierarchical nested models with different numbers of latent variables do not lend themselves well to nested model comparison [47]. Specific fit indices for the model comparisons of the non-nested higher-order models mentioned above are used (PNFI, AIC, BIC). However, CFA_1b showed the best descriptive fit (χ²(85) = 144.44, *p* < 0.001, χ²/df = 1.699, RMSEA = 0.075, SRMR = 0.067, CFI = 0.908), the lowest AIC (214.44) and the lowest BIC (313.15) compared to the other models (see Appendix A). The differences between the PNFIs of CFA_0 and CFA_1 is 0.075 and the PNFI difference of CFA_0 and CFA_2 is 0.071, which can be considered a substantial difference according to Williams and Holahan [66]. Furthermore, the CFA_1 shows the lowest CAIC (348.15) and the lowest ECVI (1.743; CI: 1.505–2.046) (for full results, see Table 3).

It should be mentioned that a negative residual variance of the latent factor FRI is observed in all hierarchical models. For pragmatic reasons, this value was initially replaced by the smallest possible value of 0.001 [68]. For further implications, see the Discussion section of this report. The resolved variance for the *g*-factor and factor-loadings from g onto EF are reported in the following section. In order to validate this finding, the model (CFA_1) was tested for common method variance.

In order to validate the preliminary findings, the best fitting model (CFA_1) was tested for common method variance. For this purpose, the factor loadings of the method factor on the indicators were freely estimated. The fit of each of the four proposed models is displayed in Table 4.

When an unmeasured latent method factor was modeled (CFA1_cmf), the fit indices improved. The likelihood-ratio test revealed a significantly better fit of CFA1_cmf compared to CFA_1b (Δχ²(14) = 23.95, *p* = 0.05). A comparison between the estimated parameters of the model without and the model with the method factor showed that lower parameter values were estimated by controlling for the method effect (see Appendix A). Accordingly, the *g*-factor in the model with an included method factor showed higher variance (σ² = 0.636; *p* = 0.000) than without the method factor (σ² = 0.188; *p* = 0.02). The squared multiple correlation (SMC) of the latent factor EF decreased from R^2^_cfa_1b_ = 0.182 to R^2^_cfa1_cmf_ = 0.078 while controlling for the method factor. If the trait/method model (CFA1_cmf) fits better than the trait-only model (cmf_to), there is evidence that trait-based and method variance are present in the data [69]. The likelihood-ratio test revealed a significantly better fit of CFA1_cmf compared to cmf_to (Δχ²(14) = 78.14, *p* ≤ 0.000). Accordingly, a common method variance (CMV) can be assumed. 

The fit of the model with the exclusive consideration of trait factors was on the cusp of acceptance. The detailed measures for assessing the fit of the initial data behaved accordingly. The average indicator reliability (squared multiple correlations) showed a value of 0.71 and fulfilled the criteria of Bagozzi and Yi [70] (≥0.60). However, four of the 15 indicator reliabilities showed a value ≤ 0.40 and thus fell below the minimum criterion [46], p. 150. The trait-only model (cmf_to) fits the data better than the method-only model (cmf_mo). The likelihood-ratio test also revealed a significantly better fit of cmf_to compared to cmf_mo (Δχ² (14) = 383.815, *p* ≤ 0.000). This could be interpreted as evidence for the observed variance in the independent and dependent constructs being not only dependent on the method [69]. In sum, the improved fit measures, as well as the significant ∆χ2 statistic for the model with trait and method factor (CFA1_cmf), suggested that common method variance was present in the data. Overall, 11 of 15 of the cross-charges were statistically significant at the 5% level of significance. The standardized loadings ranged from −0.078 to 0.642. The indicator variance explained by the method factor is approximately 21.7% on average and corresponds to variance explanations by a method factor found in the literature [71,72]. 

In order to identify any systematic effects, three additional models were specified with different equality restrictions for the cross-loadings on the method factor (for the results, see Table 4). A significant deterioration of the model fit showed that neither a homogeneous nor a construct- or scale format-specific method effect existed. Overall, a moderate common method variance could be assumed (Appendix A). 

According to a significant heterogeneous method influence that could be demonstrated with the help of the CFA, the following section examines whether this also results in a bias in the structural equation model, i.e., a common method bias. A comparison between the estimated parameters of the model with and the method factor was realized (see Appendix A). For the factors whose error variances were fixed due to having negative error variances, controlling for the method effect lower parameter values were estimated. The comparison between a modified trait/method model (R_cmf) in which factor loadings were constrained to the value obtained from the trait-only model (cmf_to) with the unmodified trait/method model (CFA1_cmf) provided information on whether a bias is present in the data [69]. There was no significant distinction between the two models and therefore no bias is assumed (R_cmf χ²(99) = 154.3, *p* ≤ 0.000, CFA1_cmf χ²(71) = 120.49, *p* ≤ 0.000, Δχ²(28) = 33.81, *p* ≤ 0.250, approximate *p* = 0.216). 

## 4. Discussion

The main aim of the current research was to investigate the common structural relationship between intelligence and EF. Contrary to previous approaches, which mainly measured EF using experimental tasks with children, the current study uses deficits in everyday EF behavior assessed by parents to examine whether primary intelligence factors and EF can be unified under a common *g*-factor. EF was proposed as being another primary intelligence factor in a comprehensive second-order model. Comparing the model fits of first- and second-order CFAs revealed that the inclusion of a second-order *g*-factor in the model does not decisively influence the fit. Regarding the appropriate fit indices, the comparison between the first-order and second-order models did not lead to a clear solution. Considering further indices of the information-theoretic measures, which take small sample sizes into account [64], an advantage of the second-order model can be assumed. A higher-order model (CFA-2a/b) in which additional indicators and further latent factors were included did not improve on the less complex model CFA_1a/b. Due to the small differences between the fit indices of the first-order and second-order CFA, it is important to discuss whether the assumption of a *g*-factor is reasonable at all [73]. Furthermore, the authors show that the inclusion of a second-order *g*-factor comes at the price of methodological impurities and does not lead to an improved fit. Considering neurobiological and genetic bases, the assumption of a common *g*-factor related to EF and intelligence seems appropriate [20,74]. Therefore, the orientation on the second-order intelligence structure model seems suitable, which has already been confirmed as part of the standardization and validity studies of the WPPSI-IV [39,40].

In the preferred models, the variance resolution of the *g*-factor for EF is 18.2% (7.5% when a method factor is taken into account). These values are remarkably lower than those obtained using directly assessed data from children using experimental EF tasks [6,9]. However, this approach was chosen to integrate the fact that, at the moment, empirical data is rarely available that focus on the deficits in everyday EF behavior from the parental perspective in relation to *g*-factor in this age range of average developed children without severe impairments. An exception is the study by Rahbari and Vaillancourt [29], where the strongest correlations were found between the working memory scale (WM) from the BRIEF-P and the subtests receptive vocabulary, expressive vocabulary, and block design of the WPPSI-III. The current study confirms the dominant correlations between the WPPSI-IV subtests and the WM scale of the BRIEF-P, and in particular, the relationship between receptive vocabulary and the WM (see Appendix A). It should be mentioned that the BRIEF-P specifically captures attentional processes that maintain information in memory but does not include updating (meant as manipulation of this content). In this regard, the working memory construct of the BRIEF-P represents only a partial aspect of updating. These processes are described as executive control in the model of Miyake and colleagues [3] but are not further distinguished in their model.

Finally, this study attempts to highlight the influence of different assessment methods. The results suggest that common method variance is present in the current data and that it influences the indicator variance by approximately 22%. This corresponds to variance explanations found via a method factor in the literature [71,72]. The question of whether this heterogeneous method effect can also be interpreted as method bias must be answered negatively. The current results indicate that EF can probably be understood as a primary factor under a *g*-factor, though this is true to a lesser extent than previous research has indicated. However, the different distribution among the indicators leads to the assumption that other effects besides the common method variance are also captured by the method factor (CMV) [67]. Thus, social desirability might influence the two methods presented here differently—in the WPPSI-IV it might be present as an increased willingness from children to exert effort and in the BRIEF-P it might be a higher willingness from parents to agree, which is due to the polarity leading to higher ratings of abnormal behavior. Analyses of the negativity and consistency scales of the BRIEF-P support the assumption of heterogeneity of CMV [42]. However, an analysis of these two scales shows no evidence of significant biases on the group level of the parent-assessed EF (Appendix A). 

Do the methodological differences between survey metrics matter for the structural equation of intelligence and EF? The use of a common method variance approach can only provide an approximate value in terms of accuracy of fit to the underlying question. 

A principal component analysis based on the data of the German standardization and validation studies confirmed a three-factor model of the BRIEF-P, consistent with the English version [42,43]. In order to avoid multiple inclusion of individual items of the three composite indices of the BRIEF-P (ISCI, FI, EMI) in this study, a modified factor model of the clinical scales is considered [24]. This multiple item inclusion could also be a reason why unidimensional models [23,24,25] as well as second-order models [25,26] with different numbers of factors affect the structure of the BRIEF-P depending on the sample used.

As noted above, the first-generation criteria for verifying CFA prerequisites were not properly implemented. Since no other sample was available for pretesting, this reliability determination was performed post-hoc using the same sample.

The same applies to the second-generation test criteria determined with the CFA [47]. Despite this simplification, these values can provide clues to the underlying measurement model. All factor reliabilities are above the required cut-off criteria (≥ 0.6) and the AVE of all proposed factors are above the cut-off (≥0.5) recommended by Weiber and Mühlhaus [47]. At least the indicator reliabilities can be considered sufficient (cut-off ≥ 0.4), except for the WPPSI-IV subtest OA (Object Assembly) and SH (Shift, BRIEF-P-scale). For detailed information, see Appendix A. To ensure the integrity of the overall model and, if possible, to fully capture the included diagnostic instruments, both parameters were left in the measurement model without adjustment.

There are high correlations within the scales of the BRIEF-P, which, on the one hand, can be explained by the theoretical framework and concurring with the model. On the other hand, items are merged several times into different indices in some cases. However, this gives rise to the risk of multicollinearity, which complicates the interpretation of the statistical model and makes the statements on the estimated parameters less precise [24]. For that reason, the BRIEF-P indices were not used as primary factors as mentioned above.

The moderate violations of requirements (normal distribution) are still within acceptable limits according to Weiber and Mühlhaus [47] and Bollen [75]. Indeed, as mentioned above, high test scores in cognitive abilities are unlikely to be overestimated. It can be assumed that a rigorous exclusion of outliers would lead to a decrease in the observed correlations.

Willoughby and colleagues [76] report a 9% proportion of 3- to 5-year-old children with saliences in their EF, referring to an epidemiological study (N= 1120). In the underlying study, 4.0–8.9% of the values related to the clinical scales of the BRIEF-P are to be considered as abnormal behavior, while for the GEC of the BRIEF-P only eight children (6.5%) were found to have noticeable deficits in EF.

Notwithstanding, the current study shows some insightful correlations that should be mentioned for future research (Appendix A). First, we want to highlight the relevance of receptive and expressive language skills in relation to executive functions. Unfortunately, the WPPSI-IV subtests picture naming (PN, expressive ability) and receptive vocabulary (RV, receptive ability) were not included in the final chosen model due to a missing improvement of the model fit at higher complexity.

Often, studies focus on receptive language performance as a parallel development between receptive language performance and EF [38,77]. Thus, the verbal cognitive abilities of the WPPSI-IV are in relation to WM and EF as well [78,79]. These findings are of important relevance, because in this age range, precursor skills turn into verbal action planning and control, the manipulation of memory content, and rehearsal strategies that are used for learning.

It seems logical that children with better language skills can regulate and direct their behavior better [80] in the sense of verbal self-instruction, even in situations in which inhibitory control seems to be effective [81,82]. Another relevant aspect is the interaction of fluid cognitive abilities with those of EF (see introduction). In this study, the highest correlations were found between the scales fluid reasoning (FRI) and inhibition (INH), confirming this assumption as well. In particular, the BRIEF-P scales show the highest correlations with the WPPSI-IV subtest picture concepts, with small to medium effect sizes.

The intercorrelation between the scales inhibition and working memory of the BRIEF-P reached a correlation coefficient r = 0.656, *p* < 0.000. Thus, the working memory and inhibit scale of the BRIEF-P appear to be closely interrelated and operate as a unit [3,6]. This corresponds to the viewpoint of some researchers [22] that EF are still largely undifferentiated in their early development. Brydges and colleagues [9] assume, on the basis of a latent variable model, that in 7- to 9-year-old children, g_f_ and g_c_ share 80% and 69% common variance with EF, respectively. Given these findings, it can be assumed that the correlations found to stem from both g_c_ and g_f_. Based on the findings of studies on the development of EF and the development of working memory capacities, we would expect this correlation to increase with age [22]. As mentioned above, the development of both working memory processes and inhibitory control from preschool age onwards is shaped by biological maturation processes [17,18]. These two areas of the EF can predict later school performance [38,76]. Thus, they play a central role in school success.

Deficits in EF, as well as in intelligence, are predictive of later academic achievement. For this reason, it is useful to investigate the predictive power of both constructs in future studies using a common model of cognitive abilities with SEM (structure equation modeling).

The results of our study illustrate the importance of examining EF at the preschool level since minor deficits in the area of EF are often covered behind the main symptoms of other developmental disorders, in particular specific language development disorders (SSED) [83] and developmental disorders with ADHD symptoms [84]. There is a need to implement screening for EF as early as possible in development, starting at kindergarten age, before the phase of accelerated central nervous system development. Due to standardized questionnaires, such as the BRIEF-P, it will be possible to identify the impairments associated with EF even before the child enters school.

Adequate support could prevent problematic school careers or decrease secondary impairments to be expected within the framework of school entry [85]. Strengthening maternal and paternal parental skills should also be brought into focus in this context [86]. A positive parenting style, characterized by responsiveness, reciprocity, and reciprocal focus, shows a positive influence on attention, problem-solving skills, and positive social behavior, benefiting younger children with ADHD symptoms in particular [87]. Additionally, professional and case-based support should be provided to educators in childcare settings. A number of behavioral therapy interventions, which are also mentioned as ADHD treatments [88], may also have a positive impact on EF, in particular on the ability to self-regulate [89]. Zelazo, Blair, and Willoughby [90] provide an overview of training interventions that decrease deficits in EF in their report for the National Center for Education Research in the United States.

In order to achieve greater sustainability and increase teaching effectiveness, combining educational and therapeutic interventions with accompanying parental counseling and behavioral therapy approaches should be considered. Additionally, the socioeconomic situation of the concerned families must be taken into account, since this is clearly related to the presence of deficits in EF [78,91]. Moreover, with the high prevalence of the symptomatology associated with EF deficits, there is increased importance attached to state and federal policies. From this perspective, it is desirable to assess EF deficits as early as kindergarten age and it should play an integral part in the initial medical examination for school readiness.

In addition to the relevance of our findings, further limitations of the results should also be mentioned. The design of this study is not suitable for answering further differentiated aspects, such as the question of the ongoing development of EF in late childhood and adolescence. The limitation of only measuring once and using a narrow age range offers only a selective view on the relation between intelligence and EF in the transition phase from preschool to school entry. A longitudinal study design will be needed to determine the developmentally relevant causal relationship between EF and cognitive abilities. Future studies could collect data from toddlerhood through middle childhood to capture a broad developmental spectrum.

There are some more limitations regarding the sample size. This study does not achieve a sufficiently large sample to provide a satisfactory EFA to reanalyze the factor structure of the instruments involved.

In addition, a small sample is also more likely to have methodological issues. In particular, negative variance (i.e., Heywood cases) and few indicators per factor (<3) are more prone to non-convergence or improper CFA solutions [92]. Some of these facts are evident in this study as well.

Due to the fact that the WPPSI-IV is also geared towards test efficiency [39], it only provides two indicators per factor, which leads to potential problems due to the small sample. Negative error variances were found in the higher order models CFA_1 and CFA_2 as well (Appendix A). Nevertheless, the error variances were fixed in order to be able to work with the given hypotheses. This limits further model modifications by adding covariances of interest.

Due to the methodological restrictions, an exploratory factor analysis (EFA) and confirmatory factor analysis (CFA) on the same sample were not realized [47] and the dimensionality of the assessed constructs using exploratory factor analysis (EFA) was not recalculated in the current work. The psychometric properties of the proposed joint model can only be estimated by relying on previous EFAs from the standardization and validation studies [39,42].

Whether a different analysis technique for the explanation of CMV would yield different results, even with the inclusion of these specific BRIEF-P scales, was not investigated in this study. Alternative approaches for controlling the methodological influences are the marker variable technique [93] or a multitrait–multimethod approach (MTMM) [66]. Due to the circumstances of our study, which is without an a priori usable marker variable, and acceptance of high feature dispersion due to the relatively small sample (in acceptance of a small number of remaining outliers), we decided against pursuing this. On the other hand, the single latent method factor approach cannot identify the specific cause of CMV [66,94].

Finally, EF was measured by parents’ ratings on the BRIEF-P. Although this standardized inventory is useful for identifying EF impairments, performance-based EF tasks may address different EF abilities than behavioral assessments [95]. Cognitive, performance-based tasks capture the efficiency of performance in an optimal environment, while inventories such as the BRIEF-P provide information about rational goal pursuit behavior in everyday situations [96]. While performance-based tasks are administered in a structured, novel, quiet, and one-on-one testing environment that is not representative of everyday life, it can be assumed that the BRIEF-P assessment is more sensitive to detecting everyday EF deficits [48].

In conclusion, our study examines normally developed children without severe impairments with respect to the integral relationship between intelligence and EF. The results indicate stable correlations at the construct level and that a superordinate and common *g*-factor is able to map deficits in EF in everyday situations to a significant extent. In this respect, these results confirm previous findings that found stable associations between intelligence and EF also in everyday situations in preschool children. This underlines the importance of an early and joint assessment of these cognitive ability domains.

## Figures and Tables

**Table 1 children-09-01089-t001:** Sociodemographic characteristics of age group sample 4 years 0 months–6 years 11 months.

	Male	Percent	Female	Percent	Sig.
Sex	75	60.5	49	39.5	n.s.
Migration background	29	38.7	19	38.8	n.s.
Parental education	n	percent	n	percent	
low	4	5.3	5	10.2	n.s.
medium	23	30.7	16	32.7	
high highest	16 32	21.3 42.7	13 15	26.5 30.6	

Note. Parental education assessed by the highest level of education achieved by either one parent (low educational level = no school-leaving qualification, at least 9 school years; medium educational level = at least 10 years at school and school-leaving qualification (“mittlere Reife” as a German education degree); high educational level = at least 11 school years, university entrance requirement; highest educational level = university degree). Migration background is indicated when either the child or at least one parent was not born in Germany; n.s. = not significant.

**Table 2 children-09-01089-t002:** Model fit indices of the tested models.

Model	Description	Goodness-of-Fit-Index							Enhancement		
		df	χ^2^	CFI	SRMR	RMSEA	AIC	BIC	Δχ2	Δdf	*p*
Independence Model		105	738.082								
CFA_0	Common model with six primary factors	75	140.53	0.898	0.0652	0.84	230.53	357.45	597.55	30	0.00
CFA1_1a	Six primary factors One secondary factor *g*-loading onto EF fixed to 0	86	161.35	0.881	0.1263	0.75	214.44	313.50			
CFA_1b	Six primary factors One secondary factor *g*-loading onto EF fixed to 1	85	144.44	0.908	0.0666	0.75	214.44	313.15	16.91	1	0.000
CFA_2a	Six primary factors Two between factors (vci) One secondary factor *g*-loading onto EF fixed to 0	114	199.56	0.881	0.1264	0.78	277.56	387.55			
CFA_2b	Six primary factors Two between factors (vci) One secondary factor *g*-loading onto EF fixed to 1	113	182.43	0.904	0.0726	0.71	262.43	375.25	18.16	1	0.000

Note: df = degrees of freedom; CFI = comparative fit index; SRMR = standardized root mean square residual; RMSEA = root mean squared error of approximation; AIC = Akaike information criterion; BIC = Bayesian information criterion; CFA_0= common first-order CFA; CFA_1a = second-order model, loading *g* on EF fixed to zero; CFA_1b = second-order model, loading *g* on EF fixed to one; CFA_2a = second-order model with two nested subfactors, loading *g* on EF fixed to zero; CFA_2b = second-order model with two nested subfactors, loading *g* on EF fixed to one.

**Table 3 children-09-01089-t003:** Fit indices of parsimony-adjusted measures and information-theoretic measures.

Model	PNFI	AIC	BIC	CAIC	ECVI	ECVI-CI
CFA_0	0.580	230.533	357.446	402.446	1.874	(1.634–2.178)
CFA_1	0.653	214.440	313.150	348.150	1.743	(1.505–2.046)
CFA_2	0.651	263.096	374.596	414.596	2.211	(1.933–2.556)

Note: Parsimony normed fit index (PNFI), Akaike information criterion (AIC), Bayesian information criterion (BIC), consistent Akaike information criterion (CAIC), expected cross-validation index (ECVI), ECVI confidence interval (ECVI-CI).

**Table 4 children-09-01089-t004:** Model fit indices of the model comparisons for common method variance with a method factor.

Model	Description	Goodness-of-Fit-Index							Chi-Square Difference Tests		
		df	χ^2^	CFI	SRMR	RMSEA	AIC	BIC	Δχ2	Δdf	*p*
CFA1	Six primary factors One secondary factor *g*-loading onto EF fixed to 1	85	144.44	0.908	0.0666	0.75	214.44	313.15			
cmf_to	Only trait factor	87	198.63	0.922	0.1056	0.75	218.08	353.45			
cmf_mo	Only meth factor	105	571.88	0.275	0.1535	0.190	601.88	644.182	383.8 ^a^	18	0.000
CFA1_cmf	Six primary factors One secondary factor free *g*-loading onto EF one method factor	71	120.49	0.923	0.0585	0.75	218.50	356.69	78.14 ^a^	16	0.000
cmf_ae	Loadings on meth factor all equal	86	197.59	0.922	0.105	0.103	265.59	361.48	75.51 ^b^	15	0.000
cmf_se	Loadings equal by scale format	85	161.56	0.881	0.881	0.086	231.56	330.27	41.07 ^b^	14	0.000
cmf_ie	Loadings equal by indicator	81	151.01	0.891	0.891	0.084	229.01	339.00	30.52 ^b^	10	0.000

Note: CFA1 = CFA_1b; trait-only model = cmf_to; method-only model = cmf_mo; cross-loadings all equal = cmf_ae; cross-loadings equal by scale format = cmf_se; crossloadings equal by indicator = cmf_ie; ^a^ comparison model: “trait factors only”; ^b^ comparison model: “cross-loading on method factor freely estimated”.

## Data Availability

The data presented in this study are available on request from the corresponding author. The data are not publicly available due to original informed consent provisions.

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
