# Peer review of "Structural Equation Modeling of Common Cognitive Abilities in Preschool-Aged Children Using WPPSI-IV and BRIEF-P"

_children, 2022, doi:10.3390/children9071089_

Round 1
Reviewer 1 Report
Introduction:
This study reports on a pre-PhD secondary analysis of data collected from data originally collected for the validation and standardisation of the WPPSI IV in Germany. The aim of the study is to understand the relationship between a direct measure of intelligence (WPPSI IV) against a parent reported measure of executive function in children aged 4 – 6 yrs 11 mo. The relationship between these two measures is studied by using confirmatory factor analysis approaches where the underlying construct is “general intelligence” (g).
The authors describe the conceptualisation of executive function and its core competencies (Cognitive flexibility, Inhibition Control, Working Memory) and make it clear that areas of executive function described as “hot components” e.g constructs of; experience of reward and punishment, control of social desires and decision making involving personal and emotional interpretation – are not their focus in this paper.
The authors describe previous research linking executive function to a central process of working memory (visual, episodic and phonological) and describe studies looking at the biological/neuroanatomical basis for this work. They do not make it entirely clear that this work has been solely conducted in adults but I assume this to be the case in reading between the lines?
The authors go on to describe the concept of intelligence and the maturation of EF and intelligence. They describe how the core EF processes occur at different times and stages. The authors describe how much EF and intelligence are interlinked. I am not entirely clear from reading the paper, but I think the authors are then describing how the research describing the linkage between EF and intelligence in this developmental period of 2-7 years is controversial. This could be spelled out better. The language used is hard to read possibly due to difficulties with writing in English. The authors go on to describe the Cattell Horn Carroll model which assumes an understanding from the reader that I was not clear on. The author describes some controversial literature which links EF to working memory but does not describe why this research is not definitive and what the gaps are in this research that lead the authors to need to write this paper. This could be much clearer for the reader.
The authors then go on to describe how EF is linked to fluid intelligence and again, I think are describing that this literature is controversial – but this is not entirely apparent in reading the paper – I think due to difficulties in writing in English.
The authors finally describe how no studies have conducted a study using confirmatory factor analysis. This could be clearer as to why this is important and what this approach will add to the literature in a way that can be understood by a reader who may know less about this subject. Can the authors explain in simpler terms why a common structural model might be useful?
This study may have some importance but it is not immediately clear to me and may need to be better spelled out to a reader who is not embroiled in a detailed debate of the factor structure of intelligence vs executive function. It may be quite important but it is not easy to read between the lines of a very densely written piece of work. Are the novel factors that it is being done in children? That is is using Confirmatory Factor Analysis? And that it is using parent reported measures of executive function? Can this all be much more spelled out and clearer?
Methods:
The authors describe their subjects but are not clear as to why the numbers are limited, whether they did a sample size calculation before commencing the study and why there are disparities in the number of girls vs boys in the study. I suspect this was for convenience for the pre-PhD work but if this is the case, it should be enunciated and discussed in the limitations. This needs to be clearer and more up front.
There is also no description of how these children are defined as “normally developing” and whether they excluded or included certain children. It is also not clear as to why certain children had a BRIEF and others not. This may bias the findings considerably and needs to be mentioned and clarified.
The authors describe the sample but do not describe how they measured socioeconomic status. Table 1 header is not entirely clear and should be able to be read “on its own” without any other text. Some of the sub headings in the table should be capitalised and generally, could be defined more easily in the table rather in the footer if there was room.
The authors then go on to describe the WPPSI IV and the BRIEF – this is quite long and detailed and I wonder if all this information needs to be presented in the paper.
The authors use the term “behaviourally conspicuous” – is this a term used in the BRIEF? I found this an odd term to use.
The authors go on to describe the use of a Confirmatory Factor Analysis. I am not an expert on this approach and would ask the editors to ensure that a statistician with this area of expertise review the paper from this perspective. From a reader’s perspective who is not an expert in this area, I would benefit from an easier description of why this supports the work undertaken. There is also some description of the management of “outliers” which uses a lot of very detailed opaque language – I wonder if again, some of this could be simplified for the average reader?
Results and discussion.
The data and tables presented are quite clear although I was surprised that the authors might not consider at least placing one of the six factor model figures in the main text – could they also explain better what CMV is?
The interpretations and conclusions are justified but quite difficult to read due to the English writing style. The authors describe the fact that only one measure of intelligence was needed rather than a secondary measure. The authors describe other literature but they do not entirely clarify why their study may have either been biased and not fit into this literature or why their model may have meant that they got different results. Again, I think some of this comes down to difficulties in interpreting the discussion due to the writing style.
The paper is generally very difficult to read and very dense and long with an assumption of a very detailed knowledge on factor analysis. In reading between the lines, the paper may have merit but it really needs someone who speaks English as a first language to rewrite the text so that it is understandable. It also should be shortened and more to the point and I would hope that the authors could simplify some of the language or explain it better to a reader who may be in the field of psychology and child development/executive function but who needs some of the analysis clarified. A good scientist should be able to ensure that their text is clear and transparent and I think some of this may be just an issue of English language writing.
I have highlighted some text with some ideas about changing the English language but i think it will need quite a bit more to get it to a readable level.

Author Response
We would like to thank you for your constructive and detailed input on our current research paper. Additionally, we are very grateful for the comments you gave in the provided pdf file. Below we have outlined our corrections or explanations in accordance to the chronology of your recommendations.
We did some major revisions concerning the English editing in all sections of the paper. First, we made basic revisions by ourselves including content specifications (For example, we changed “behaviourally conspicuous” into “abnormal behavior”). Second, we used the English pre-editing-service by MDPI. Therefore, the editor accepted all previous changes and then made his/her revisions using “track changes”.
We attached three files for transparency: 1. _revised_before editing, 2. _english edited and the final version 3. _revised final.
We believe to have successfully incorporated your feedback concerning unclear wording into our revised manuscript. For instance, we included a sentence what the CHC-model is meant for, highlighted that research in preschool-aged children is missing so far, and tried to disentangle the literature more stringent (working memory, fluid intelligence, methodological approaches, aim of the study).
Regarding the methods, we clarified the selection of the sample, excluding was only by methodological issues, as already mentioned in the paper.
In this study, we did not calculate explicitly the socioeconomic status as for example described by Lamperts and colleagues. Therefore, we already included a table with information about migration and parental education with more details in the notes. We decided to not include this additional information in the table, because we think this makes it harder to focus on the values. We did some graphical and spelling revisions of the table about sociodemographic characteristics and hope, that it is more readable now.
Yes, we know the parts about the WPPSI IV and the BRIEF are quite long. We decided to not compress these descriptions because of the methodological focus of the study. The same applies to the section of the statistical methods. In both sections we also have done major revisions, so, we hope, it is more simplified now.
In the results, we inserted what CMV means and made some minor revisions in spelling.
In the discussion we also done major revisions in language, we think this leads to a more stringent interpretation of the current findings and tried to more relate them to the literature.
Overall, we shortened the sentences and brought them more to the point. Therefore, the language should be clearer and transparent now.

Reviewer 2 Report
The study is very interessant. The main purpose it was investigate the common variance to executive functions and intelligence, defined as general intelligence (g), in which g represents variance common to mental tests and arises from ubiquitous positive correlations among tests. The positive correlations indicate that people who perform well on one test generally perform well on all others. Interessant also is that Executive Functions based on parent rating were measured for indicates the cognitive deficits of children in the everyday behavior.
The experiment and their hypotheses are very well conduced and very well described and analyzed. I appreciated very much the results presented as well the discussion made considering all results obtained. . The study should be a very important contribution for the are.
Author Response
We would like to thank you very much for your review. We have done major revisions regarding the language and minor revisions based on recommendations of the second reviewer.
Round 2
Reviewer 1 Report
The authors have done some major revisions concerning the English language within the paper. It reads better but it still does not flow naturally and is very dense for the average reader. The research could be publishable but in its current format, it is really difficult to understand and assumes a very specific knowledge that might be better in a psychometric journal rather than this journal – unless it is made much clearer. The introduction is quite “clunky” and difficult to follow and needs a rewrite and more clarity as to what points the author is making and why so it can flow from one part to another. This is not just specific spelling or grammar issues, but it needs someone with a better writing style to revise it.
The authors were asked to complete a major revision however most of what has been revised is the English language but little else. I am concerned that the authors have not addressed many of the points relating to the methodology and the explanation of the methodology which I think would need to be addressed before the paper reaches a stage that it would be fit for publication.
Even this first sentence in the abstract; “…is a unitary construct in which the subordinate factors; “Updating”, “Inhibition” and “Shifting” ….” is extremely confusing and does not make sense. Are the authors trying to say that Research evidence to date has demonstrated that executive function with it’s subordinate parts (Updating, Inhibition and Shifting) only correlates moderately with a general factor of intelligence”? Are they therefore trying to say that there is a gap in the literature? This is not clear as the authors only state that evidence already shows the two are not strongly correlated – maybe they want to explain how their study is filling this gap?? I am not clear at all as to what the authors want to say. Would the author’s writing style be better with a more “active voice” to clarify some of these statements?
In my previous review, I requested that the authors provide more information on the “normal sample” that they utilized for this study. It is not clear as to how they have justified that they have a “normal sample” and what inclusion and exclusion criteria ensure that this is a “normal sample”. This needs to be clarified. A random sample was taken of children from different kindergartens, day care centres and pre-school centres but it is not clear as to whether there were any characteristics of these settings that were considered beforehand e.g. are these sites representative of the whole of Germany in terms of sociodemographics and if not, what are the differences?
The authors assume that a reader will understand (in the abstract) what a “second order solution” is. Do you think the readership will understand this or should it be clarified in the abstract somehow?
The results are still unclear in the abstract and the conclusion in the abstract is only that results are discussed in terms of “current research findings and further practical implications”. This seems to me “wishy washy” and not very clear and pointed.
Is it EF or EFs? If it is EF then it should follow with “has” not “have” and is should be “plays” rather than “play” – please check this throughout.
Is it “Inhibition Control” or “Inhibitory control”? I have not heard the term “Inhibition Control” before but this may be right. Might be worth checking
I had previously asked whether the authors might explain better what “the neuropsychological perspective described by Raiford and Coalson [30], and based on along with the Luria model and clinical experience, provides information suggests that EF are integral aspects of cognitive performance in the re-spective subtests and indices of the WPPSI-IV and are not hierarchically structured within them“. This sentence does not make sense and i think it should be clear what the Luria model and the perspective from Raiford and Coalson is.
Author Response
We would like to thank you again for your detailed input on our current research paper. Below we have outlined our corrections or explanations in accordance to the chronology of your recommendations.
We implemented some more language corrections to make our paper more readable. Due to some complex relations between singular aspects, some of the phrasings remain complex. We hope, that the mainly text is better to understand now.
Regarding the aspect, that it should be more readable for the average reader and that it assumes specific statistical knowledge, we think that our recent paper matches well to one aspect of the focus of research in intelligence, which is the theme of this special issue. We hope that it addresses to some interested researchers.
Regarding the introduction, we tried to rewrite some parts, so it is clearer now, what the authors intend and how they investigated this.
In our first major revision, we focused on language issues and therefore made just small revisions in methodology. Now we tried to revise this section in principle and one the one hand, cut some too detailed explanations and on the other hand, added some more simplified formulations.
Thank you again for your helpful comment, that our statement in the abstract is not clear. We now included the sentence “Furthermore, we intend to close the gap of more daily life associated executive functions and replicate findings in preschool aged children.”, hoping that it clarifies our intention.
We now implemented more information about the “normal” sample, inclusion and exclusion criteria and comparison with the sample of the standardization study. Because it does not differs from this sample, that is representative for German preschoolers, we assume the representativeness of our
Especially, we changed the expression socioeconomic characteristics into sociodemografic characteristics, reduced the headings in table 1 and included ages in the text.
Regarding your question, if the reader will already understand in the abstract what a second order solution means, I also think, that you are right. Due to limitations in the number of words in the abstract, we hope that this will be clearer for the readership, when the methodology and result section is read. We think, that it could be helpful for researchers to include this statistical expression in the abstract, if they are searching for results.
Your point that the sentence included in the abstract “The results are discussed in terms of current research findings and further practical implications.” says not really anything, might be true. It is only the description of what follows in the paper, because it is very complex and would take too much space in the abstract. We can delete this sentence, if you think that it is unnecessary.
We corrected again the grammatically issues about singular or plural and abbreviation of Executive Functions and hope we detected all errors now. Usually Executive Functions are abbreviated as EF, but are ment to be the plural.
Thank you for your comment on the expression Inhibitory Control vs. Inhibition Control. We now corrected this.
Regarding your constructive comment that the sentence on the neuropsychological perspective, that we tried to link with the WPPSI, is not clear, we decided to delete this part, because it would take much more space to get deeper into this aspect.
Sincerly